

# Evaluating a linkage between obesity and the occurrence of dental caries among school going children in Sakaka, Al Jouf, Kingdom of Saudi Arabia

Osama Khattak[1], Azhar Iqbal[1], Farooq Ahmad Chaudhary[2], Jamaluddin Syed[3,4], Thani Alsharari[5], Sudhakar Vundavalli[6], Bayan Abdullah Sadiq Aljahdali[7], Ahmed Eidan Abdullah AlZahrani[8], Rakhi Issrani[6] and Sherif Elsayed Sultan[9,10]

[1] Department of Restorative Dentistry, Jouf University, Sakaka, Saudi Arabia
[2] School of Dentistry (SOD), Federal Medical Teaching Institution (FMTI)/PIMS, Shaheed Zulfiqar Ali Bhutto Medical University (SZABMU), Islamabad, Pakistan
[3] Oral Basic Clinical Sciences Department, Faculty of Dentistry, King AbdulAziz University, Jeddah, Saudi Arabia
[4] Department of Regenerative Dentistry, University of Bari, Bari, Italy
[5] Restorative and Dental Materials Department, Faculty of Dentistry, Taif University, Taif, Saudi Arabia
[6] Department of Preventive Dentistry, College of Dentistry, Jouf University, Sakaka, Saudi Arabia
[7] General Dentist, Arar, Saudi Arabia
[8] General Dentist, Taif, Saudi Arabia
[9] Department of Fixed Prosthodontics, Tanta University, Tanta, Egypt
[10] Department of Prosthetic Dentistry, Jouf University, Sakaka, Saudi Arabia

Corresponding authors
Osama Khattak,
dr.osama.khattak@jodent.org
Farooq Ahmad Chaudhary,
chaudhary4@hotmail.com

## ABSTRACT

**Background.** Obesity and dental caries are global public health problems. There are conflicting reports about the relationship between caries and obesity. The aim of this study was to analyze the type of relationship between the dental caries and obesity among school children in Al-Jouf region of Saudi Arabia.

**Methods.** This cross-sectional study was conducted among 400 participants aged 6 to 14 years. The study involved measuring caries (dmft/DMFT), assessing body mass index (BMI), and administering a self-completion questionnaire. An independent $t$-test, one-way ANOVA, and multivariate logistic regression analyses were performed.

**Results.** Out of 400 participants, 380 agreed to participate in the study. Overall caries prevalence among the participants was 76.1% and mean DMFT and dmft values were 2.8 ± 1.0 and 3.7 ± 1.6. Among the factors associated with mean caries scores, relation between DMFT scores and frequency of consumption of sugar was statistically significant (F = 3.82,0.01). Regression models has identified children with increased BMI values has 3.2 times more risk of getting dental caries in permanent teeth ($P$ = 0.001).

**Conclusion.** There was a positive association between obesity and dental caries in school going Saudi Arabian children. Comprehensive multidisciplinary approach by health professionals is recommended for preventive public health issues related to caries and obesity in teenagers.

## INTRODUCTION

Numerous changes have been observed over the last few decades in the diet and lifestyle of the general population due to rapid urbanization, industrialization, and economic development (*Swinburn et al., 2011*). These changes include consumption of high amounts of carbohydrates and lower physical activity, which negatively impact their general and dental health, especially in young age groups, causing dental caries and obesity (*Fernandez et al., 2017*). These two conditions are considered chronic, widespread, and multifactorial, having a devastating impact on the lives of children and youth (*Fernandez et al., 2017*).

Dental caries have become a global public health issue for pre-school children and lead to pain, chewing difficulties, and disorders of general health affecting the growth and development of children (*Fernandez et al., 2017*; *Hayden et al., 2013*). Studies from various regions of the world reported the prevalence of dental caries in children ranges from 40–90%, and in Saudi Arabia its prevalence is reported to be around 80% in primary and 70% in permanent dentition (*Al Agili, 2013*; *Fernandez et al., 2017*; *Tsai, Hsiang & Johnsen, 2000*).

Similar to dental caries, prevalence of obesity is also very high and it is becoming one of the most common general health problems among school children (*Akpata, Al-Shammery & Saeed, 1992*). A cross-sectional study in America reported the prevalence of obesity among children of age 4 at 18%, in Taiwan among 3-10 years children it was in the range of 4% to 17%, and in Saudi Arabia obesity has become an emerging public health problem among the children due to its alarming high rate (*Al Othaimeen, Nozha & Osman, 2007*; *Anderson & Whitaker, 2009*; *Tsai, Hsiang & Johnsen, 2000*). If obesity remains uncontrolled, it can lead to devastating health conditions such as high blood pressure, diabetes, stroke, and cardiovascular disease (World Health Organization). All these conditions ultimately can lead to disabilities and death (*Sturm, 2002*). Studies have shown that obesity and dental caries have become common childhood diseases that are affecting the overall health and development of children (*Wake et al., 2007*). Many contributing factors are common for both these conditions; these include lifestyle, biological, dietary, genetic, cultural, socioeconomic, and environmental factors (*Chi, Luu & Chu, 2017*; *Spiegel & Palmer, 2012*). Therefore an association between dental caries and obesity seems logical and knowledge about this relationship is very important so that more effective and targeted public measures could be initiated to reduce its prevalence (*Chi, Luu & Chu, 2017*). It is also believed that understanding more about this relationship can help to control both of these conditions at once (*Scorzetti et al., 2013*). Previous studies that examined this relationship provided conflicting findings and the relationship between obesity and dental caries is still unclear (*Fernandez et al., 2017*; *Scorzetti et al., 2013*).

No such study has been conducted in the Al-Jouf region of Saudi Arabia; therefore, no data is available about the association between dental caries and obesity in the school children in this region. Hence, the present study was designed to examine and report the type of relationship between dental caries and obesity among school children in the Al-Jouf region of Saudi Arabia.

## MATERIALS & METHODS

This cross-sectional study was carried out between March to May 2021 among the children of Sakaka, Al-Jouf province, Saudi Arabia. The children aged between 6-14 years old and permanent residents of the Sakaka region were selected for this study using stratified cluster random sampling method. Initially schools were selected through simple random (lottery) method from list of the schools in Sakaka. From the selected schools children were selected using systematic random sampling method. Ethical approval was obtained from the Local Committee of Bioethics, Jouf University (approval no. LCBE/07-06-43). The children who were medically compromised and had any psychological disorders were excluded from this study.

The sample size calculation was based on the prevalence of dental caries reported by *Gudipaneni et al. (2021)*. For this investigation, separate sample sizes were determined for distinct caries prevalence rates for different BMI groups, with the largest calculated sample size among all being used. Considering 95% confidence levels, and 80% power, 350 children were sufficient to detect the clinically significant difference of 10% in the prevalence of caries. The total sample size was adjusted to 400, expecting the fact that there were approximately 10–15% chances of non-participation/rejection.

Participation was voluntary, both parents and children were informed about the purpose of the study before taking oral and written consent from children and either of the parents. It was assured that the results of the study would be only presented or published as aggregate data maintaining the confidentiality of personal information.

All the selected children underwent intraoral examination by two trained and calibrated dentists. Dental caries was recorded using dmft/DMFT (decayed, missing, filled teeth for deciduous and permanent dentition) based on WHO methods for oral health surveys (*World Health Organization, 1987*), DMFT for permanent teeth and dmft for deciduous teeth were recorded separately. For the calibration process the inter- and intra-examiner reliability was analysed using Cohen's kappa coefficient that was ranging from 0.78 to 0.82 indicates fair to a good agreement.

The body mass index (BMI) was used to evaluate obesity among children by calculating their weight in kg divided by the square of height in meters (kg/m2). The weight of the children was recorded on a physician's scale and a stadiometer was used to measure the height. The BMI scores of children were categorized according to the cut-off values described by International Obesity Task Force, as follows: BMI group- Moderate malnutrition (BMI 16–16.99), Mild malnutrition (BMI 17–18.49), Normal (BMI 18.5–24.99), Overweight (BMI 25–29), Obese (BMI 30≥) (*James, 2004*).

The self-administered questionnaire collected information regarding demographic characteristics (age), dietary (sugar consumption), and oral hygiene habits (frequency of tooth brushing, use of dental floss, and mouthwash/ rinsing habits). Quantification was done on tooth brushing frequency and the amount of sugar consumed/day. Sugar consumption was determined from 24-hour dietary recall chart filled by parents on two different days with one week gap and average of both charts was taken as sugar consumption

**Table 1  Participant characteristics ($N = 380$).**

| Gender | N (%) |
|---|---|
| Boys | 211 (55.5%) |
| Girls | 169 (44.5%) |
| Total | 380 |
| **Age (years)** | |
| 6 | 24 (6.3%) |
| 7 | 31 (8.9%) |
| 8 | 42 (11%) |
| 9 | 62 (16.3%) |
| 10 | 32 (8.4%) |
| 11 | 41 (10.6%) |
| 12 | 62 (16.1%) |
| 13 | 44 (11.4%) |
| 14 | 42 (11%) |
| Mean age | 9.8 Years |
| Range | (6–14 years) |
| **BMI Scores** | **Mean (Range)** |
| Male | 28.2 (17–34) |
| Female | 24.3 (16–30) |
| Overall | 26.4 (16–34) |

value for the participant. Face-to-face interviews of 20 parents were conducted to assess the reliability of the questionnaire and Cronbach's alpha of 0.75 was obtained in this study.

## Statistical analysis

The analysis of collected data was carried out in IBM SPSS software version 24.0 with the significance level set at 5%. The comparison of mean dental caries scores with gender was examined using an independent sample $t$-test and the relationship of mean caries scores with BMI scores was analysed using Pearson's correlation coefficient. Multiple regression was performed for caries scores as a dependent variable and various independent variables which were significant in bivariate analysis.

## RESULTS

Out of 400 children contacted, 380 children were consented to take part in this study with a response rate of 95%. The mean age of the sample was 9.8 years, and it was consisted of 211 (55.5%) males and 169 (44.5%) female participants (Table 1).

Mean dmft/DMFT values (dmft = 4.2 and DMFT = 3.2) were highest among the children who didn't brush their teeth regularly compared to the children who brushed regularly which was non-significant statistically. A minor difference was reported in mean dmft/DMFT values between the children who floss regular and who didn't floss *i.e.,* dmft of 3.8 *vs* 3.6 and DMFT of 2.4 *vs* 2.2. Children who used mouthwash/who have mouth rinsing habits had fewer caries in permanent dentition (2 *vs* 2.8) and deciduous dentition (3.9 *vs* 3.2). Mean DMFT values were directly proportional to the frequency of intake of sucrose

**Table 2  Comparison between dental caries, oral hygiene practices and dietary habits of participants.**

| | Mean dmft/DMFT scores | | Caries prevalence (dmft/DMFT ≥1) |
|---|---|---|---|
| | Deciduous | Permanent | |
| **Gender** | | | |
| Male | 3.8 ± 1.2 | 3 ± 1.3 | 79.4% |
| Female | 3.6 ± 1.4 | 2.7 ± 0.8 | 72.6% |
| Overall | 3.7 ± 1.6 | 2.8 ± 1.0 | 76.1% |
| Unpaired 't' test (t value) | 1.54 | 3.22 | |
| P-value | 0.69 | 0.04* | |
| **Brush teeth per day** | | | |
| Less than once | 4.2 ± 1.8 | 3.2 ± 1.6 | 81.3% |
| Once a day | 3.4 ± 0.8 | 2.2 ± 1 | 73.2% |
| Twice a day | 2.1 ± 1.2 | 2 ± 1.2 | 76.5% |
| More than Twice a day | 1.9 ± 0.9 | 1.8 ± 1.4 | 75.1% |
| ANOVA (F value) | 1.29 | 0.51 | |
| P-value | 0.27 | 0.72 | |
| **Dental floss** | | | |
| Yes | 3.8 ± 1.4 | 2.4 ± 1.8 | 74.8% |
| No | 3.6 ± 1.6 | 2.2 ± 1.4 | 78.5% |
| Unpaired 't' test (t value) | −0.92 | −0.62 | |
| P- value | 0.43 | 0.53 | |
| **Mouthwash/rinsing habit** | | | |
| Yes | 3.2 ± 1.2 | 2.0 ± 0.8 | 75% |
| No | 3.9 ± 1.4 | 2.8 ± 1.2 | 77.3% |
| Unpaired 't' test | 0.51 | 1.24 | |
| P- value | 0.82 | 0.07 | |
| **Frequency of consumption of Sucrose containing foods** | | | |
| Once a day | 3.4 ± 2.1 | 2 ± 0.8 | 66.4% |
| 2–4 times a day | 3.2 ± 1.8 | 2.2 ± 1.2 | 70% |
| 4–6 times a day | 2.8 ± 1.2 | 2.3 ± 1.6 | 74.2% |
| More than 6 times a day | 3.8 ± 1.0 | 2.6 ± 1.8 | 81.3% |
| ANOVA (F value) | 0.99 | 3.82 | |
| P-value | 0.23 | 0.01* | |

**Notes.**
SD, Standard deviation.
*Statistically significant.

diet in permanent dentition, mean DMFT of 2 in children who consumed sucrose diet once a day to 2.6 in people who consumed sucrose diet more than 6 times a day. Table 2 presents the distribution of mean caries scores (dmft/DMFT) according to oral hygiene habits, and sugar consumption.

BMI values of the children were compared to mean caries scores as shown in Table 3. A statistically significant association was found between overweight (mean DMFT = 2.8)

**Table 3  Relationship between BMI and dental caries experience among study participants.**

| | BMI group | | | | | ANOVA | |
| | | | | | | F value | P value |
| | Moderate malnutrition | Mild malnutrition | Normal | Over weight | Obese | | |
|---|---|---|---|---|---|---|---|
| Mean dmft | 3.4 ± 2.1 | 3 ± 1.6 | 3.6 ± 1.8 | 3.6 ± 2.2 | 4.8 ± 2.8 | 1.23 | 0.08 |
| Mean DMFT | 2 ± 1.1 | 2.4 ± 1.4 | 2.4 ± 0.8 | 2.8 ± 1.6 | 3.9 ± 2.2[*] | 4.36 | 0.001[*] |
| Caries Prevalence | 64.2% | 65.6% | 67.3% | 78% | 83.6% | | |

Notes.
   *Statistically significant.

**Table 4  Multiple regression analysis to identify factors influencing DMFT score.**

| | Model | Unstandardized coefficients | | Standardized coefficients | P-value | 95% CI |
| | | b | S.E | b | | |
|---|---|---|---|---|---|---|
| 1 | (Constant) | 2.111 | .383 | – | 0.001[*] | 1.35, 2.86 |
| | Sugar intake | 0.43 | .080 | 0.42 | 0.29 | −.064, .20 |
| | BMI | 3.20 | 1.69 | 0.82 | 0.001[*] | 2.06, 3.80 |
| | Gender | 0.04 | .076 | 0.04 | 0.52 | −.10, .20 |

Notes.
   *Statistically significant.

and obesity (mean DMFT = 3.9) with caries scores in permanent dentition and also for deciduous dentition but was statistically insignificant.

Multiple regression was performed for the statistically significant independent variables observed in bivariate analysis. Gender, BMI values, and sugar intake were included in the model. The odds ratio for BMI to caries score in permanent dentition was found to be 3.2 which was statistically significant whereas other variables were not significant (Table 4).

## DISCUSSION

A plausible indirect biological association of dental caries and obesity has been argued in literature considering the fact that poor dietary habits and inappropriate diet not only promotes obesity but also create favourable condition for dental caries to flourish, especially with increased intake of sugary food (*Gerdin et al., 2008*). However further research and investigations are needed to establish this relationship due to its controversial status. Hence, this study was designed to analyze this type of relationship between caries and obesity in Saudi Arabian 6 to 14 years old school-going children.

The mean dmft/DMFT in this study was higher in those children who brushed less than once a day as compared to those who brushed once and more per day. This finding was statistically insignificant. This result was in line with the results of the previous studies where a greater risk of dental caries was related to irregular tooth brushing habits (*Ashour, Ashour & Basha, 2018*; *Chaudhary & Ahmad, 2021*; *Chaudhary, Ahmad & Bashir, 2019*; *Chaudhary, Ahmad & Sinor, 2021*; *Farooqi et al., 2015*). Additionally, other oral hygiene habits like usage of dental floss and mouthwash/rinsing habits were found to be not significant among the study participants.

On the other hand, a comparison between caries (dmft/DMFT) and dietary habits among the study participants showed that mean DMFT values were directly proportional to the frequency of intake of sucrose diet in permanent dentition. A similar finding was noted in the study done by *Ashour, Ashour & Basha (2018)* in which the subjects with sugar consumption were more likely to have caries than subjects with no sugar consumption (*Ashour, Ashour & Basha, 2018*). In the last 30 years, the Saudi Arabian population has witnessed marked changes in their lifestyle and dietary habits resulting in deteriorating health behaviours with a more sedentary lifestyle, reduce physical activity, and fat-rich diet (*Alshihri et al., 2019*).

Results of this study showed that the obese children were 3.2 times more likely to have caries than the non-obese children in their permanent dentition. The association of weight and caries in the literature still remains controversial; however, the results of this study indicated a strong relationship between overweight status and dental caries. A systematic review by *Hayden et al. (2013)* had shown a similar association of obesity and increased level of dental caries in permanent dentition (*Hayden et al., 2013*). A previous study in the Al-Khobar region of Saudi Arabia also showed a similar association between obesity and caries experience (*Al-Ansari & Nazir, 2020*). On the other hand, according to *Ravelomantsoa, Razanamihaja & Randrianarivony (2019)* who had conducted a review based on cross-sectional studies published between 2010 and 2015 found that dental caries were associated with high and low body mass index (*Ravelomantsoa, Razanamihaja & Randrianarivony, 2019*).

In this study, dental caries in the primary dentition showed no correlation with the BMI scores, and similar results were found in another study conducted by *Hayden et al. (2013)*, suggesting that older children will be more likely to be obese due to increasingly sedentary lifestyle with age (*Hayden et al., 2013*).

Few studies have reported an inverse association between caries and BMI (*Cheng et al., 2019*; *Fernandez et al., 2017*; *Kopycka-Kedzierawski et al., 2008*). However, the mechanism of this inverse relationship is difficult to explain as it is evident from the literature that consumption of sugary food and refined carbohydrates contributes to the development of dental caries and obesity (*Fernandez et al., 2017*). A plausible argument that can support this inverse association can be attributable to the dietary patterns of obese. People with obesity do not necessarily consume more sugary and refined carbs diet rather they consume more fatty, fried, and unrefined carbohydrates foods. This dietary pattern could certainly increase obesity, but not necessarily effects dental caries (*Alshihri et al., 2019*).

The other perception related to the diet is that people who frequently consume snacks have a greater risk of obesity and caries, but many studies that examine this relationship of snacking habits with the weight status found either no relationship or reported opposite relationship where young people who consumed snacks in between the meals were less likely to be obese (*Fernandez et al., 2017*). Apart from the diet itself the process of mastication is greatly affected by caries which can lead to feeding difficulties and nutritional intake deficiency mostly in children and older people (*Alshihri et al., 2019*). A study by *Gilchrist et al. (2015)* showed that children with untreated caries had difficulties in chewing hard food along with food getting stuck in teeth which restricted their practice of eating a

healthy diet (*Gilchrist et al., 2015*). The consequences of these dietary limitations are not just limited to weight only but would reduce consumption of important dietary nutrients such as calcium, phosphate, and vitamins A and D, which are essential for developing and maintaining healthy teeth and gums (*Alshihri et al., 2019*). Another aspect of an inverse association of snacking with dental caries is the increased secretion of saliva due to more food consumption by obese people that could eventually reduce the incidence of caries owing to its protective effect. Additionally, studies have mentioned other factors such as poverty and low socioeconomic status that explain the relationship of being underweight and having more dental caries (*Chaudhary & Ahmad, 2021*; *Chaudhary, Ahmad & Bashir, 2019*; *Farsi et al., 2016*; *Kumar et al., 2017*).

Few studies have reported that BMI did not correlate with caries, such as studies in Saudi Arabia and Brazil on school-going children showed no association between BMI and dental caries in both primary and permanent dentition (*Alves et al., 2013*; *Da Silva et al., 2016*). Similarly, a recent review paper that aimed at investigating the association between obesity and dental caries in young in England had reported that obesity was neither associated with prevalence nor with severity of dental caries. They also found that the association between dental caries and obesity was moderated by the effect of deprivation, white ethnicity, and lone parenthood (*Ravaghi et al., 2020*). In a systematic review of caries and adiposity reported a similar finding of no relationship between overweight and dental caries that was assessed by BMI (*Kantovitz et al., 2006*). One possible explanation for this no correlation might be due to the multifactorial etiological nature of both dental caries and obesity that are impacted by various genetic and environmental factors (*Alshihri et al., 2019*).

Some studies showed a positive association between BMI and caries; however, this relationship is stronger and more reliable in permanent than in primary dentition (*Al-Ansari & Nazir, 2020*). The burden of caries and obesity usually increases with age as these diseases gradually accumulate during the life course, therefore a stronger association is observed with an increase in age as a burden of these diseases also increases with age. In this study similar pattern was observed as obesity was statistically significantly associated with dental caries in permanent teeth. There are some common risk factors mentioned in the literature that can increase the likelihood of both obesity and caries (*Hayden et al., 2013*). These risk factors can either be modifiable such as behavioural and psychological factors or can be non-modifiable such as biological, genetic, socio-demographic, and cultural (*Ashour, Ashour & Basha, 2018*; *Gerdin et al., 2008*; *Sakeenabi, Swamy & Mohammed, 2012*).

Higher sugar consumption is one of the common examples of behavioural modifiable factors that negatively affect both conditions (*Hayden et al., 2013*). Therefore, the research focusing on dental caries and obesity should include the risk factors and use the integrating and collaborating strategies in prevention of it both locally and globally. The cross-sectional studies design which we used in this study is not suitable for establishing the causal relationship. However, this type of study can provide very important information regarding the prevalence, risk indicators and provide useful data that can encourage further longitudinal studies and help in monitoring the oral health conditions of Saudi Arabian children.
## CONCLUSION

In this study, a significant association was found between obesity and caries experience among school children of Al-Jouf, KSA. This finding provides important information for the prevention and management of dental caries in children by focusing on specific risk factors associated with these two diseases having common risk determinants. There is a need for a comprehensive and integrated effort by both medical and dental healthcare professionals for the prevention and management of obesity and caries in children by implementing practical solutions. Future research should investigate the specific factors of overweight that can play the protective role against dental caries in permanent dentition.

### Funding

This research received funding from Jouf University Saudi Arabia (reference number DSR 2020-04-2579). The funders had no role in study design, data collection and analysis, decision to publish, or preparation of the manuscript.

### Grant Disclosures

The following grant information was disclosed by the authors:
Jouf University Saudi Arabia (reference number DSR 2020-04-2579).

### Competing Interests

The authors declare there are no competing interests.

### Author Contributions

- Osama Khattak conceived and designed the experiments, performed the experiments, authored or reviewed drafts of the article, and approved the final draft.
- Azhar Iqbal conceived and designed the experiments, prepared figures and/or tables, and approved the final draft.
- Farooq Ahmad Chaudhary conceived and designed the experiments, prepared figures and/or tables, and approved the final draft.
- Jamaluddin Syed performed the experiments, prepared figures and/or tables, and approved the final draft.
- Thani Alsharari performed the experiments, prepared figures and/or tables, and approved the final draft.
- Sudhakar Vundavalli performed the experiments, authored or reviewed drafts of the article, and approved the final draft.
- Bayan Abdullah Sadiq Aljahdali performed the experiments, analyzed the data, authored or reviewed drafts of the article, and approved the final draft.
- Ahmed Eidan Abdullah AlZahrani analyzed the data, prepared figures and/or tables, and approved the final draft.
- Rakhi Issrani conceived and designed the experiments, authored or reviewed drafts of the article, and approved the final draft.

- Sherif Elsayed Sultan analyzed the data, authored or reviewed drafts of the article, and approved the final draft.

## Ethics

The following information was supplied relating to ethical approvals (i.e., approving body and any reference numbers):

Ethical approval was obtained from the ethical local committee of bioethics, Jouf University (Ref: 07-06-43).

## Data Availability

The raw measurements are available in the Supplementary File.

## Supplemental Information

Supplemental information for this article can be found online at http://dx.doi.org/10.7717/peerj.13582#supplemental-information.

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
