# Peer review of "Evaluating a linkage between obesity and the occurrence of dental caries among school going children in Sakaka, Al Jouf, Kingdom of Saudi Arabia"

_PeerJ, doi:10.7717/peerj.13582_

## Round 0.1 · original submission · Major Revisions

Please double-check the manuscript for spelling/grammatical errors. Please also respond to the reviewer's comments and make necessary revisions. One of the reviewer's comments are in the attached annotated pdf.

Reviewer 1 ·

Basic reporting

it was sufficient.

Experimental design

Poor design

Validity of the findings

weak

Annotated reviews are not available for download in order to protect the identity of reviewers who chose to remain anonymous.

Reviewer 2 ·

Basic reporting

The manuscript is well written, I do not have any major comments, however few questions and suggestions i would like to mention.

1. Introduction: line 67-69 - If obesity remains uncontrolled, it can lead to devastating health conditions such as high blood pressure, diabetes, stroke, and cardiovascular disease. (Provide references here).

2. Introduction: line 70-71- Studies have shown that obesity and dental caries have become common childhood diseases that are affecting the overall health and development of children(Wake et al. 2007). (You mentioned studies, therefore provide few more references here).

Experimental design

3. Methods: Please write full ethical approval reference number and name of the approval authority.

4. Method: How do you manage the dmft/DMFT in mixed dentition which contains both deciduous and permanent teeth in different combinations? It seems that this study analyse the dmft and DMFT separately without considering the mixed dentition.

5. The authors perform regression on DMFT (permanent dentition) alone. How about dmft (deciduous dentition)? The age of the recruited subjects and their number of deciduous/permanent teeth will have impacts on the result and analysis.

6. The discussion seems elaborate and explaining all concerns related to the current evaluation. However, needed to cite few latest studies.

Validity of the findings

In this manuscript, the author ensures careful study planning, sample size and data analysis.

---

## Round 0.2 · accepted · Accept

I am satisfied with the changes made by the authors. The manuscript can be accepted in its current form.

Reviewer 1 ·

Basic reporting

Accepted

Experimental design

Accepted

Validity of the findings

Accepted

Reviewer 2 ·

Basic reporting

Revised manuscript is now improved. I do not have further comments.

Experimental design

Revised manuscript is now improved. Raised concerns are answered and modifications are made in manuscript.

Validity of the findings

In this manuscript, the author ensures careful study planning, sample size and data analysis.

Additional comments

I congratulate authors for a very well executed study. The manuscript has been revised according to the suggestion and comments.